# In Vitro Dynamic Model Evaluation of Meropenem Alone and in Combination with Avibactam Against Carbapenemase-Producing *Klebsiella pneumoniae*

**DOI:** 10.3390/ph17121683

**Published:** 2024-12-13

**Authors:** Elena N. Strukova, Yury A. Portnoy, Maria V. Golikova, Stephen H. Zinner

**Affiliations:** 1Department of Pharmacokinetics & Pharmacodynamics, Gause Institute of New Antibiotics, 11 Bolshaya Pirogovskaya Street, 119021 Moscow, Russia; strukovagause@gmail.com (E.N.S.); yaportnoy@gmail.com (Y.A.P.); 2Harvard Medical School, Department of Medicine, Mount Auburn Hospital, 330 Mount Auburn St., Cambridge, MA 02138, USA; szinner@mah.harvard.edu

**Keywords:** antibiotic resistance, in vitro hollow-fiber dynamic model, β-lactams, β-lactamase inhibitors, carbapenemase-producing *Klebsiella pneumoniae*

## Abstract

**Background:** A potential strategy to maintain the efficacy of carbapenems against carbapenemase-producing *Klebsiella pneumoniae* (CPKP) is their combination with carbapenemase inhibitors. To address these issues, the effectiveness of a novel combination of meropenem with avibactam against CPKP was studied. Additionally, the applicability of a pharmacokinetically-based approach to antibiotic/inhibitor minimum inhibitory concentration (MIC) determinations to better predict efficacy was examined. **Methods:** CPKP strains were exposed to meropenem alone or in combination with avibactam in an in vitro hollow-fiber infection model. Treatment effects were correlated with simulated antibiotic and antibiotic/inhibitor combination ratios of the area under the concentration–time curve (AUC) to the MIC (AUC/MIC). All MICs were determined at standard and at high inocula; combination MICs were determined using the conventional approach with fixed avibactam concentration or using the pharmacokinetic (PK)-based approach with a fixed meropenem-to-avibactam concentration ratio, equal to the respective drug therapeutic AUC ratios. **Results:** Meropenem alone was not effective even against a “susceptible” CPKP strain. The addition of avibactam significantly improved both meropenem MICs and its effectiveness. The effects of meropenem alone and in combination with avibactam (merged data) correlated well with AUC/MIC ratios only when MICs were determined at high inocula and using the PK-based approach (*r*^2^ 0.97); the correlation was worse with the conventional approach (*r*^2^ 0.73). **Conclusions:** The effectiveness of meropenem/avibactam against CPKP is promising. A single “effect–AUC/MIC” relationship useful for predicting meropenem efficacy (alone or in combination with avibactam) was obtained using MICs at high inocula and combination MICs determined using a PK-based approach.

## 1. Introduction

Carbapenems are first-line treatments for critically ill, infected patients [1]. However, their efficacy is gradually decreasing due to increasing prevalence of carbapenem-resistant Gram-negative bacteria that produce carbapenemases [2]. Although some strains of *Klebsiella pneumoniae* exhibit low carbapenem minimum-inhibitory concentrations (MIC), there is currently no evidence that they might not still produce carbapenemases. A clear correlation has not yet been established between a strain’s MIC and its ability to produce carbapenem-hydrolyzing enzymes. [3]. It is important to identify such strains in order to develop effective treatment strategies. Various methods are available to detect carbapenem-resistant bacteria, including the use of specialized laboratory equipment and testing kits [4,5,6]. While these resources may not always be available in clinical laboratories, it is possible to detect these bacteria by measuring their carbapenem MICs at bacterial concentrations that are significantly higher (at least 100-fold) than those used in standard testing protocols. We have previously demonstrated the applicability of this approach [7,8] and aim to further substantiate its effectiveness in this paper. High bacterial density may substantially reduce the effectiveness of carbapenems, as seen with results of susceptibility testing at both moderate and high inocula levels and in in vivo experiments in laboratory animals [9,10,11].

One way to overcome potential carbapenem resistance is to use carbapenem antibiotics in combination with carbapenemase inhibitors. Recent developments in this area have seen the introduction of several non-β-lactam β-lactamase inhibitor agents such as relebactam [12], vaborbactam [13], durlobactam [14], and avibactam [15]. Avibactam in particular is an inhibitor of class A, C, and some class D β-lactamases. Its spectrum of activity extends to KPC and OXA-48 carbapenemases, making it a valuable tool in combating antibiotic resistance [16]. When combined with ceftazidime, it provides activity against 95% of *Pseudomonas aeruginosa* isolates and 99% of *Enterobacterales*, even those that carry extended-spectrum β-lactamases (ESBLs) [17]. In this study, we evaluated the antibacterial efficacy of avibactam in combination with meropenem [18]. As far as we are aware, the pharmacodynamics of this combination have not been previously investigated, with only a limited number of promising in vitro and in vivo studies reported for this combination [19,20].

Another objective of this study is the investigation of the reliability of predicting efficacy of the meropenem–avibactam combination based on the MIC of the combination against the studied bacterial strains. One key factor influencing outcomes is the concentration of the inhibitor, which inactivates bacterial enzymes and allows the antibiotic to reach its target. The traditional approach to determine β-lactam MICs assumes the concentration of β-lactamase inhibitor to remain constant [21]. This method fails to consider the actual concentrations of antibiotic and inhibitor at the site of infection as well as their ratios. Additionally, the checkerboard assay, which also assesses drug combination activity, does not reflect drug pharmacokinetics, as it uses arbitrary antibiotic/inhibitor concentration ratios. The European Medicines Agency states that simulations should consider the variability in plasma/serum exposures to both the β-lactam and β-lactamase inhibitor as well as any potential pharmacokinetic interactions between them when they are co-administered to patients [22]. We recently introduced a pharmacokinetic-based (PK-based) approach to the determination of antibiotic/inhibitor MICs that allows for these considerations. This approach is based on the ratio of the area under the antibiotic concentration–time curve (AUC) to that of the inhibitor. In cases where the antibiotic and inhibitor have similar pharmacokinetics, the concentration ratio for these drugs is also constant. Compared to the conventional method, the use of a PK-based approach for MIC determination is feasible. We used this approach to enable more accurate predictions of the efficacy of doripenem in the presence of relebactam [23] as well as daptomycin with rifampicin [24] and linezolid with daptomycin [25]. The assumption that MICs estimated at a constant ratio of ampicillin and sulbactam can be used to predict the combination’s effect was made in 1996 by A. Firsov [26]. Another parameter—time above instantaneous MIC (where MIC is estimated at escalating but not fixed inhibitor concentration)—was used by the group of V. Tam to determine the combination exposure that provides the antibacterial effect [27,28]. Another group based their statements on ceftazidime–avibactam MICs at a fixed ratio to calculate target achievement [29]. Thus, there is interest in the development and evaluation of alternatives to traditional susceptibility testing methods, such as the PK-based approach discussed in the current study.

To evaluate the effectiveness of the novel combination of meropenem with avibactam against carbapenemase-producing *K. pneumoniae*, an in vitro hollow-fiber infection model that mimics the pharmacokinetics of meropenem and avibactam in epithelial lining fluid was used to simulate moderate and severe infections (i.e., with moderate and high bacterial loads, respectively).

## 2. Results

### 2.1. Meropenem and Meropenem/Avibactam Susceptibility Testing

The current study was conducted using three *K. pneumoniae* strains that differed in their meropenem susceptibility at a standard inoculum plus an ATCC strain as a control. The MICs varied 16-fold: 2, 8, and 32 mg/L (Table 1); the strains included variants that were susceptible or resistant to meropenem according to the EUCAST guidelines. When the inoculum was high, meropenem MICs for all strains increased at least 16-fold, suggesting the inoculum effect. When avibactam was added to meropenem at a fixed concentration of 4 mg/L, combination MICs decreased sharply at the standard inoculum by a maximum of 256-fold. At higher inocula, the addition of avibactam also enhanced the activity of meropenem, although to a lesser extent: up to a 64-fold decrease in MIC. With the control strain *K. pneumoniae* 6305, which did not contain carbapenemases, meropenem MIC was not affected by inoculum changes. All MIC values varied by no more than plus or minus one dilution, or they did not change at all.

When the meropenem-to-avibactam concentration ratio was equal to the drugs’ area under the concentration–time curve (AUC) ratio of 8/1, MICs of the meropenem/avibactam combination were mostly lower than the MIC of meropenem alone (Table 2). However, this reduction was less pronounced than with the fixed avibactam concentration, with a maximum decrease of up to 32-fold at both standard and high inocula.

### 2.2. Meropenem Pharmacodynamics Alone and in Combination with Avibactam at a Starting Inoculum of 10^6^

When meropenem was administered alone, its efficacy was dependent on the *K. pneumoniae* strain. The corresponding time–kill curves are presented in Figure 1. As seen in the figure, meropenem did not inhibit the growth of the meropenem-susceptible strain *K. pneumoniae* 561 or the meropenem-resistant strain *K. pneumoniae* 410. With *K. pneumoniae* 1905, there was some effect during the first 24 h, but subsequent bacterial counts gradually increased until reaching their maximum at 96 h. In contrast, meropenem showed bactericidal activity against the control strain *K. pneumoniae* 6305 (see Appendix A).

In the presence of avibactam, the antibacterial effect of meropenem was more pronounced with all strains. However, with *K. pneumoniae* strains 561 and 410, the combination of meropenem and avibactam was somewhat less effective due to a slight increase in cell growth.

The values of coefficient of variation (CV) for log CFU/mL data ranged from 0.7 to 26.1%. However, for most of the experiments, the CVs were <15%.

### 2.3. Meropenem Pharmacodynamics Alone and in the Presence of Avibactam at the Starting Inoculum of 10^8^

At high starting bacterial inocula, meropenem alone was ineffective against all *K. pneumoniae* strains (Figure 2) except for *K. pneumoniae* strain 6305, which served as a negative control (Appendix A). In the presence of avibactam, the effect of meropenem was restored, and the minimal bacterial counts for all three strains remained at approximately 4 log (CFU/mL).

The values of CV for log CFU/mL data ranged from 0 to 28.3%. However, for most of the experiments, the CVs were <15%.

### 2.4. “ABBC–AUC/MIC” Relationships

In the current study, we examined the effectiveness of meropenem alone and in combination with avibactam against strains of *K. pneumoniae* with varying levels of susceptibility to meropenem. Additionally, we aimed to compare the effects of the drugs at standard inoculum levels of 10^6^ CFU/mL, which simulates a moderate infection, and at higher inoculum levels of 10^8^ CFU/mL, which mimics a severe infection. To achieve this, we constructed relationships between the meropenem effect (either alone or in combination with avibactam), expressed as the area between the control growth curve and the time–kill curve (ABBC) and the respective AUC/MIC ratios. During the construction of these relationships, meropenem MIC values in the presence of avibactam were determined either using the standard method at a fixed concentration of avibactam or were evaluated using a pharmacokinetic-based ratio of meropenem and avibactam concentrations of 8/1. The results obtained from experiments with moderate inoculum load are presented in Figure 3.

As shown in Figure 3, regardless of the method used to determine the meropenem MIC in the presence of avibactam, both cases were described by a sigmoid function with a low r-square correlation coefficient of 0.36. This was actually due to a drop-down point (traced by the red circle) that corresponds to strain *K. pneumoniae* 561, which did not respond to meropenem treatment even at moderate initial inocula. Data from the experiments with the control carbapenemase-negative strain 6305 were added for comparison only. The “ABBC–AUC/MIC” relationships derived from pharmacodynamic experiments using high initial inocula are shown in Figure 4.

In all monotherapy treatments, meropenem ABBCs were low, as the antibiotic was not effective. When meropenem MIC was determined at a fixed avibactam concentration of 4 mg/L, the resulting relationship was described by a sigmoid function with an *r*^2^ of 0.73. However, when the meropenem MIC was determined using a pharmacokinetic-based approach, the relationship was described by a sigmoid function with a higher *r*^2^ of 0.97. Data points obtained from experiments with the control carbapenemase-non-producing strain 6305 are included for comparison.

## 3. Discussion

In our study, the meropenem-susceptible *K. pneumoniae* 561 strain with a meropenem MIC of 2 mg/L [30] rendered meropenem ineffective, unlike the high-dose regimen that was simulated in the hollow-fiber infection model. The lack of meropenem effectiveness suggests that if the strain is a carbapenemase producer, the antibiotic may not be effective. Recently, several clinical studies have shown that in infections caused by Gram-negative bacteria, it is preferable to use a susceptibility MIC breakpoint of 0.5 mg/L or less rather than the traditional 2 mg/L threshold [31]. In this and our previous study, we obtained similar results for carbapenem-resistant strains [32]. Unlike a high-dose meropenem regimen, which was tested previously in an in vitro dynamic model, current results indicate that the effectiveness of meropenem is not maintained against two of the three studied *K. pneumoniae* strains, even when the starting bacterial inoculum was moderate. Additionally, meropenem was not effective against high microbial counts. The combination of meropenem and avibactam demonstrated a significant enhanced effect in all experiments, reducing bacterial counts up to four-fold compared to the initial inoculum. These findings suggest that this combination potentially could be used against carbapenem-resistant *K. pneumoniae* strains.

To quantitatively evaluate the effectiveness of meropenem and to relate it to antibiotic exposure, we calculated the ABBCs and correlated them with the corresponding AUC/MIC values. For the combination regimen, the AUC/MICs were calculated using meropenem MICs in the presence of avibactam and determined either at a fixed inhibitor concentration (Figure 3a and Figure 4a) or at a fixed PK-based meropenem-to-avibactam concentration ratio of 8/1 (Figure 3b and Figure 4b). In the presence of a moderate starting bacterial load, the resulting “ABBC–AUC/MIC” relationships were described with a low r-square correlation coefficient (Figure 3). This is likely due to a drop-off in the data point for the meropenem-susceptible *K. pneumoniae* strain 561. Indeed, for this strain, meropenem did not appear to be effective even at a moderate initial bacterial load. Consequently, the susceptibility of the strain at standard inocula was not predictive of meropenem effectiveness when used alone. A possible explanation for the ability of strain *K. pneumoniae* 561 to survive at high concentrations of meropenem is a significant increase in expression of the gene encoding carbapenemase synthesis. This phenomenon of antibiotic-induced gene expression has been described in several studies, both for carbapenemase production and bacterial efflux pumps [33,34].

Obviously, a sudden increase in the ability of bacteria to resist an antibiotic that is not reflected at standard MIC levels poses a significant risk. However, this actually can be predicted based on meropenem MIC values measured at higher inoculum levels, which were significantly increased for all tested bacterial strains, including *K. pneumoniae* strain 561. It is worth noting that this is not usually inherent in carbapenemase-negative *K. pneumoniae* strains. We previously reported the potential utility of antibiotic MIC values obtained under high initial microbial loads in conjunction with standard MIC testing in predicting meropenem effectiveness [7,8]. Furthermore, the use of high inoculum MIC values for both meropenem and its combination with avibactam allows the establishment of relationships that are well described by a sigmoid function (Figure 4). However, when meropenem MIC values in the presence of avibactam (in the AUC/MIC ratio) are determined using the traditional approach (*r*^2^ 0.73), the correlation is noticeably lower compared to meropenem MICs determined using a PK-based approach (*r*^2^ 0.97). This difference suggests that the susceptibility of meropenem in the presence of avibactam determined using the PK-based method better reflects the antibiotic’s antibacterial activity than when the concentration of avibactam is fixed. This result seems reasonable, as the PK-based approach considers the actual ratio of drug concentrations at the infection site. In contrast, the traditional method fails to do so, and the ratio of antibiotic and inhibitor in the MIC assay becomes arbitrary. Our previous studies with doripenem and relebactam further support this conclusion [23]. There are some other studies that also do not support the conventional method with the fixed concentration of β-lactamase inhibitor as the appropriate way to estimate susceptibility to the β-lactam combination. Differences between MICs were noticed in a comparative study of several β-lactam/β-lactamase inhibitor combinations [35]. The results of susceptibility testing for β-lactam/β-lactamase inhibitor combinations against *Acinetobacter baumannii* depended on whether a fixed inhibitor/β-lactam ratio or a fixed concentration of the inhibitor was used; some MICs varied up to five dilutions. The authors suggested that the therapeutic outcome may depend on the actual combination MICs, which can be determined accurately only if tested at a fixed inhibitor/β-lactam ratio. V. Tam and colleagues proposed that the conventional susceptibility testing method does not determine “true” susceptibility of some isolates [27,28]. These authors established the relationship between the response of the carbapenem MIC and escalating inhibitor concentrations and identified the optimal exposures (expressed as the time above instantaneous MIC) to reach the acceptable antibacterial effect of piperacillin and ceftazidime plus tazobactam or avibactam against *K. pneumoniae* and *Escherichia coli.* The relationship between MIC estimated at a constant ratio of ampicillin to sulbactam of 2/1 with the effect of the combination allowed the prediction of the antibacterial activity of the combination in an in vitro model as early as 1996 [26]. K. Coleman and co-authors published susceptibility data obtained by the conventional method and by doubling dilutions of a fixed 4/1 ceftazidime–avibactam combination [29]. The ratio of 4/1 was determined by Sy S.K.B and colleagues [36]. The authors stated that this alternative approach would allow achievement of the critical-concentration β-lactamase inhibitor pharmacodynamic target simultaneously with achieving the β-lactam target.

Given the current results, further investigation of the potential benefits of meropenem–avibactam combinations seems warranted. Furthermore, even when the pathogen’s apparent meropenem susceptibility is close to the susceptibility breakpoint, the use of a combination of antibiotic with a carbapenemase inhibitor may be warranted.

Our study has a significant limitation: It did not include other β-lactam antibiotics or a wide range of *K. pneumoniae* strains (those producing other carbapenemases and with different susceptibilities) to establish a more reliable relationship between the effect and pharmacokinetic/pharmacodynamic indices; additional studies should be considered.

## 4. Materials and Methods

### 4.1. Antimicrobial Agents and Bacterial Strains

Meropenem powder was purchased from Sigma-Aldrich (St. Louis, MO, USA); avibactam powder was purchased from AChemBlock (Hayward, CA, USA). Three strains of *K. pneumoniae*, none of which exhibited a hypervirulent or hyperviscous phenotype and which differed in their susceptibility to meropenem, were used in the study: Two were *bla_KPC_*-positive by PCR: *K. pneumoniae* 410 and ATCC 1905; one was *blaOXA-48*-positive by PCR: *K. pneumoniae* 561; and one carbapenemase-non-producing clinical strain *K. pneumoniae* 6305 was used as negative control in MIC testing and pharmacodynamic experiments. Bacteria were grown in Mueller–Hinton broth or Mueller–Hinton agar (Becton Dickinson, Franklin Lakes, NJ, USA) and incubated at 37 °C. Carbapenemase production was monitored throughout the study for each bacterial strain using a modified carbapenem-inactivation method [37].

### 4.2. Antibiotic Susceptibility Testing

Susceptibility testing for antibiotic and inhibitor alone or in combination was performed using broth microdilution techniques with a standard inoculum of approximately 5 × 10^5^ CFU/mL (SI) and high inoculum of 5 × 10^7^ CFU/mL (HI). Single meropenem MICs at SI were determined according to standard recommendations using Mueller–Hinton broth (Becton Dickinson, Franklin Lakes, NJ, USA) [38]. When the MICs (for single meropenem and for its combination with avibactam) were determined at HI, bacterial growth was quantified by optical density at 600 nm (OD), and ODs before and after 18 h incubation at 37 °C were estimated and verified by quantitative subculture [39]. The MIC was defined as the dilution at which the 18 h OD was equal to or less than that at time 0. The inoculum effect was defined as an eight-fold or greater increase in MIC when tested with HI compared to that for SI [9]. For meropenem/avibactam combinations, MIC testing was performed according to traditional approach (by method 1) or PK-based approach (method 2) regarding the meropenem-to-avibactam therapeutic AUC ratio. Before reading, plates were incubated at 37 °C for 18 h. MIC values were obtained at least in triplicate, and the modal MICs were estimated.

A meropenem susceptibility MIC breakpoint equal to 2 mg/L according to EUCAST recommendations was used [21].

Method 1 (traditional approach, MIC1 [21]): MIC testing for meropenem/avibactam combination used a fixed avibactam concentration of 4 mg/L with doubling dilutions of meropenem.

Method 2 (PK-based approach, MIC2): MIC testing for meropenem/avibactam combinations used a fixed PK-based carbapenem-to-avibactam concentration ratio of 8/1 by varying the carbapenem and avibactam concentrations in parallel in each subsequent dilution. These concentration ratios are equal to the 24-h area under the concentration–time curve (AUC) ratios of meropenem [40] to avibactam [41] simulated in a hollow-fiber infection model. An 8/1 meropenem-to-avibactam concentration ratio in MIC testing corresponds to the therapeutic 8/1 AUC ratio of meropenem (for a 2000 mg dose of every 8 h) to avibactam (for a 500 mg dose every 8 h): 350 mg × h/L of meropenem to 42 mg × h/L of avibactam, 350/42 = 8/1.

### 4.3. In Vitro Dynamic Model

A previously described two-compartment in vitro model (a hollow-fiber infection model) [42] was used in pharmacodynamic simulations of single and combined treatments with meropenem and avibactam. Briefly, the model consists of three connected cameras: one containing fresh cation-supplemented Mueller–Hinton broth (CSMHB), which supplied CSMHB to the second camera—the central unit used for drug dosing—and the third camera—a hollow-fiber bioreactor (Fresenius dialyzer, model AV400S, Fresenius Medical Care AG, Bad Homburg Germany)—was a peripheral unit used for bacterial cultivation, representing the infection site. The central unit and bioreactor were connected, and continuous exchange of CSMHB between these units by peristaltic pump (Cole-Parmer Instrument Company, Masterflex L/S 07523-80, Vernon Hills, IL, USA) provided maintenance of target drug concentrations in both cameras.

The operational procedure used in the pharmacodynamic experiments was as described elsewhere [42]. Each experiment was performed at least in duplicate. Antibiotic dosing and sampling were processed automatically using computer-assisted controls. The system was filled with sterile CSMHB and placed in an incubator at 37 °C. The hollow-fiber bioreactor was inoculated with an 18 h culture of *K. pneumoniae* at appropriate cell concentration. After a 2 h period of incubation, the resulting exponentially growing bacteria reached ~5 × 10^5^–10^6^ (SI) or ~5 × 10^7^–10^8^ (HI) colony-forming units (CFU)/mL. Then, the antibiotic or the antibiotic with inhibitor was administered into the central unit of the model. The duration of each experiment was 120 h. The reliability of the pharmacokinetic simulations was confirmed earlier [23].

### 4.4. Antibiotic Dosing Regimens and Simulated Pharmacokinetic Profiles

The pharmacokinetic profiles of meropenem and avibactam in epithelial lung fluid (ELF) were reproduced in vitro using a linear, one-compartment model. For this purpose, data were utilized to determine the drug concentrations in ELF obtained from healthy volunteers following the administration of 2000 mg of meropenem as a 2-h intravenous (IV) infusion every 8 h and 500 mg of avibactam as a 2-h IV infusion every 8 h [40,41]. The calculated pharmacokinetic parameters used to replicate the concentration-versus-time profiles in the in vitro dynamic model of meropenem and avibactam were as follows: maximum concentration (Cmax) values of 38 micrograms per milliliter (μg/mL) and 5 μg/mL, respectively; time to maximum concentration (Tmax) of 2 h and half-life (t1/2) of 1.5 h for both drugs; and AUC values of approximately 350 mg/L × h and 42 mg/L × h, respectively. With all dosing regimens, a series of mono-exponential profiles that mimic a thrice-daily dosing of meropenem and avibactam were simulated for 5 consecutive days.

### 4.5. Quantitation of the Antimicrobial Effect

In each experiment, the bacteria-containing medium from the hollow-fiber bioreactor was sampled to determine bacterial, antibiotic, and inhibitor concentrations throughout the observation period. Samples (100 µL) were serially diluted as appropriate, and 100 µL was plated onto Mueller–Hinton agar plates that then were placed in an incubator at 37 °C for 24 h. The lower limit of accurate detection was 1 × 10^2^ CFU/mL (equivalent to 10 colonies per plate).

Based on the time–kill data, the area between the control growth curve and each time–kill curve (ABBC) [43] was determined from the beginning of the treatment to 120 h.

Relationships between AUC/MIC (using MIC1 and MIC2) and the meropenem effect alone or in the presence of avibactam (ABBC) were described with the sigmoid (Equation (1)) function:*Y* = *a*/{1 + exp [−(*x* − *x*_0_)/*b*]}(1)
where *Y* is ABBC, *x* is log (AUC/MIC), *a* is the maximal value of the ABBC, *x*_0_ is x corresponding to *a*/2, and *b* is a parameter reflecting sigmoidicity.

All calculations were performed using SigmaPlot 12 software (Systat Software Inc., headquartered in San Jose, CA, USA).

### 4.6. Statistical Analysis

The reported MIC data were obtained by calculation of the respective modal values.

In pharmacodynamic experiments, the bacterial count data were calculated as arithmetic means ± standard deviations for three replicate experiments. Based on these data, kinetic growth and time–kill curves were constructed. Assuming that the coefficient of variation of the log CFU/mL data was less than 15%, in order to facilitate figure viewing, we decided not to include data point error bars in order to avoid interference with the kinetic curves.

## 5. Conclusions

The following findings were derived from the study: (1) Meropenem was ineffective against a carbapenemase-producing strain with an MIC of 2 mg/L; (2) an inoculum effect on meropenem MIC values was observed with carbapenemase-producing strains but not with those that did not produce carbapenemases, which limits the specificity of the phenomenon to carbapenem-producing strains; (3) the addition of avibactam significantly improved both the meropenem MICs and its effectiveness; (4) meropenem effectiveness (both alone and in combination with avibactam) could be predicted in an in vitro dynamic model based on AUC/MIC values calculated using MICs with increased inoculum; and (5) assessment of the meropenem–avibactam combination MIC was more accurate when performed using the pharmacokinetic (PK)-based approach compared with the standard method.

## Figures and Tables

**Figure 1 pharmaceuticals-17-01683-f001:**
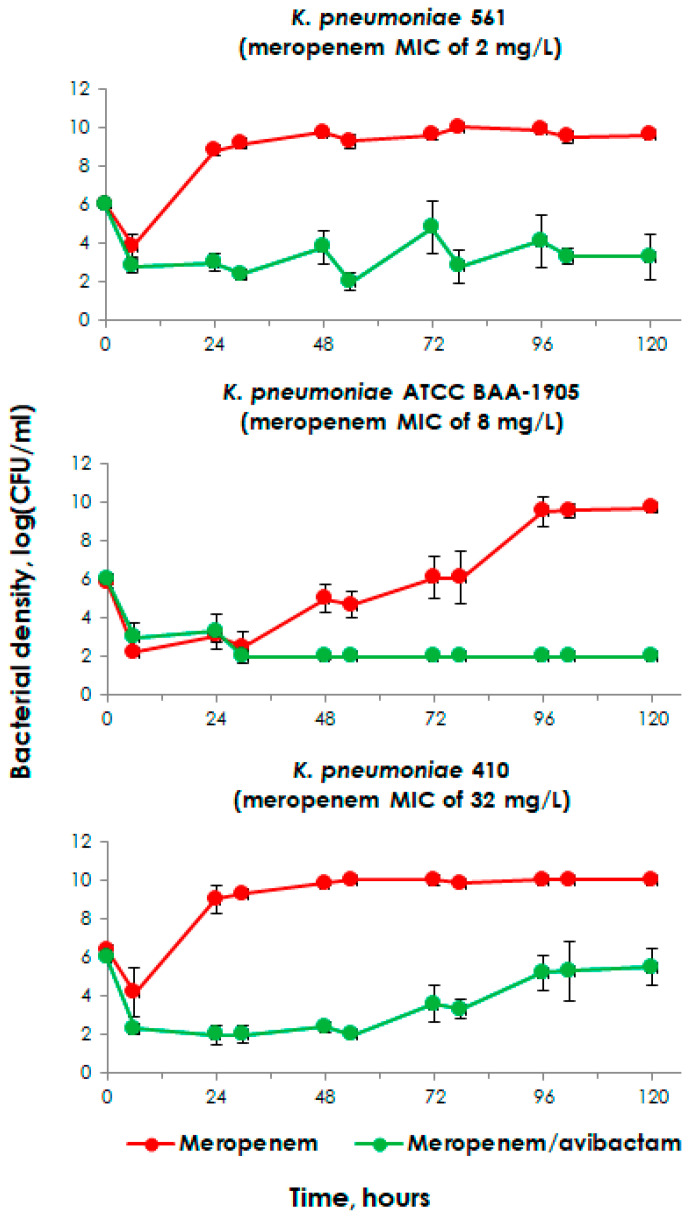
Time–kill curves of *K. pneumoniae* exposed to meropenem alone or in combination with avibactam in pharmacodynamic experiments with moderate starting inocula. The represented data were obtained from three replicate experiments. The error bars demonstrate the variability of the results.

**Figure 2 pharmaceuticals-17-01683-f002:**
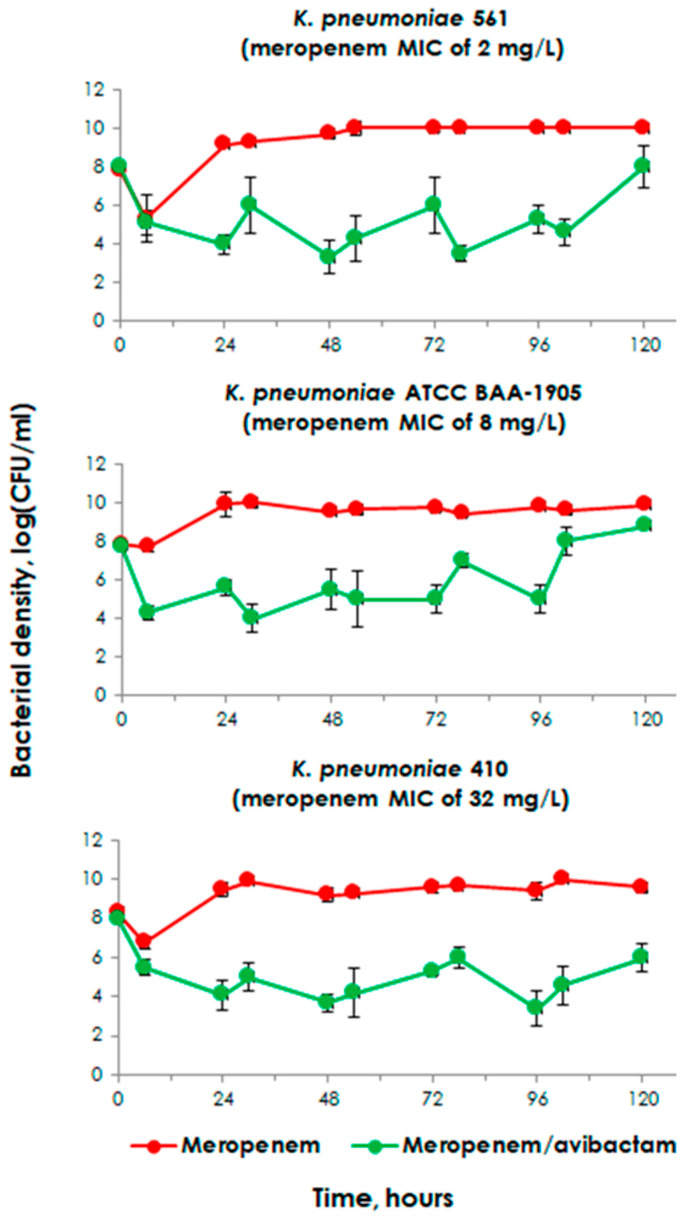
Time–kill curves of *K. pneumoniae* exposed to meropenem alone or in combination with avibactam in pharmacodynamic experiments with high starting inocula. The represented data were obtained from three replicate experiments. The error bars demonstrate the variability of the results.

**Figure 3 pharmaceuticals-17-01683-f003:**
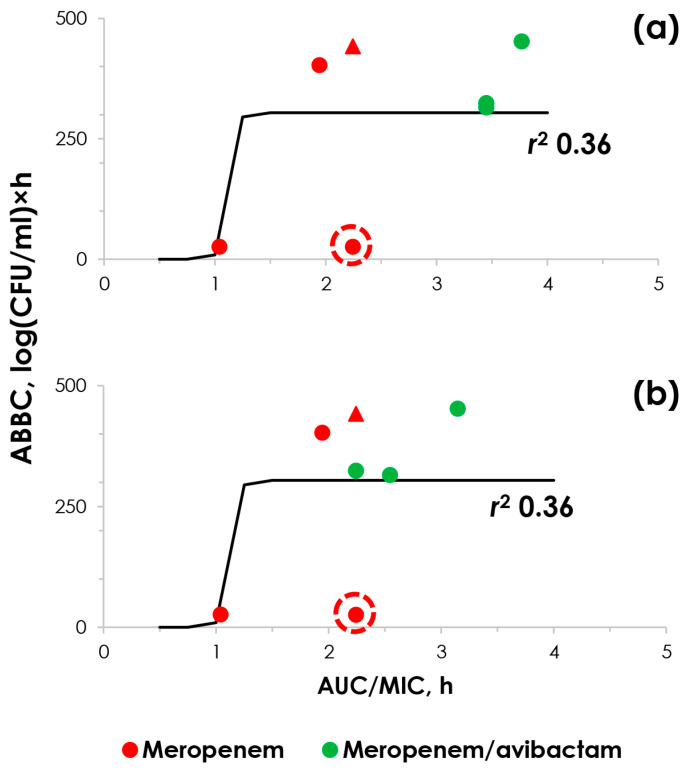
“ABBC–AUC/MIC” relationships obtained at moderate *K. pneumoniae* inocula. (**a**) Meropenem MICs determined at a fixed avibactam concentration of 4 mg/L were used for combination AUC/MIC calculations; (**b**) meropenem MICs determined at the PK-based meropenem-to-avibactam concentration ratio were used for combination AUC/MIC calculations. Red triangle—experiment with carbapenemase-non-producing *K. pneumoniae* strain 6305. The point indicated by the red circle corresponds to the meropenem-susceptible *K. pneumoniae* strain 561, with a MIC of 2 μg/mL.

**Figure 4 pharmaceuticals-17-01683-f004:**
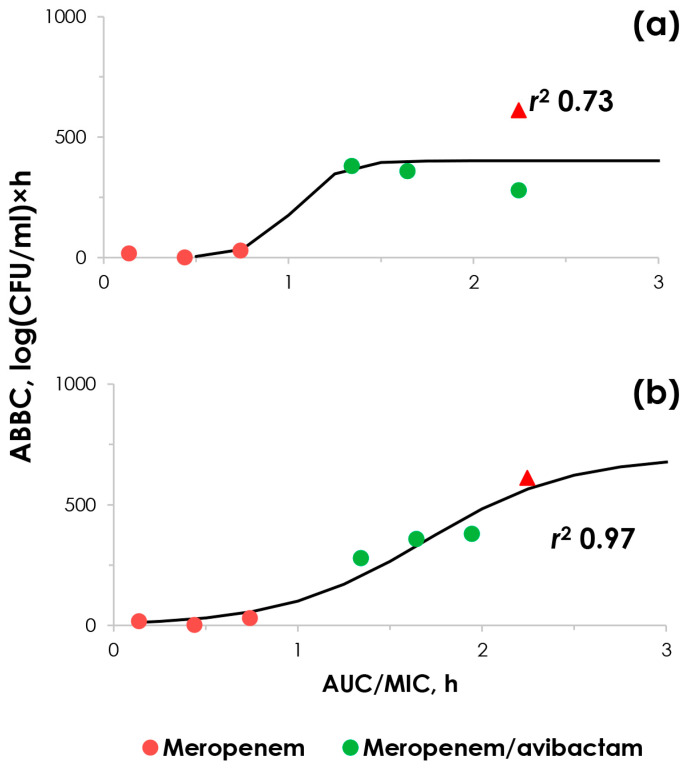
“ABBC–AUC/MIC” relationships obtained at high *K. pneumoniae* inocula. (**a**) Meropenem MICs determined at a fixed avibactam concentration of 4 mg/L were used for combination AUC/MIC calculations; (**b**) meropenem MICs determined at the PK-based meropenem-to-avibactam concentration ratio were used for combination AUC/MIC calculations. Red triangle—obtained in experiments with carbapenemase-non-producing *K. pneumoniae* strain 6305.

**Table 1 pharmaceuticals-17-01683-t001:** Susceptibility testing of *K. pneumoniae* strains to meropenem alone and in the presence of avibactam at fixed concentration of 4 mg/L.

*K. pneumoniae* Strain	Standard Inocula			High Inocula		
	Meropenem MIC, µg/mL	Meropenem/Avibactam MIC, µg/mL	x-Fold MIC Decrease	Meropenem MIC, µg/mL	Meropenem/Avibactam MIC, µg/mL	x-Fold MIC Decrease ^2^
561	2	0.125	16 ^2^	64 (32 ^1^) *	8	8
1905	8	0.06	132 ^2^	128 (16 ^1^) *	2	64
410	32	0.125	256 ^2^	256 (8 ^1^) *	16	16
6305	2	ND	ND	2 (1 ^1^)	ND	ND

* indicates when the inoculum effect was observed; ND—not determined; ^1^ indicates a inoculum-related x-fold decrease in meropenem MIC; ^2^ indicates x-fold decrease in meropenem MIC in the presence of avibactam.

**Table 2 pharmaceuticals-17-01683-t002:** Susceptibility testing of *K. pneumoniae* strains to meropenem alone and in the presence of avibactam at the pharmacokinetic-based concentration ratio of 8/1.

*K. pneumoniae* Strain	Standard Inocula			High Inocula		
	Meropenem MIC, µg/mL	Meropenem/Avibactam MIC, µg/mL	x-Fold MIC Decrease	Meropenem MIC, µg/mL	Meropenem/Avibactam MIC, µg/mL	x-Fold MIC Decrease ^1^
561	2	2	1 ^1^	64	8	8
1905	8	0.25	32 ^1^	128	16	8
410	32	1	32 ^1^	256	4	64

^1^ indicates x-fold decrease in meropenem MIC in the presence of avibactam.

## Data Availability

Data are contained within the article and Appendix A.

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
