# Peer review of "In Vitro Dynamic Model Evaluation of Meropenem Alone and in Combination with Avibactam Against Carbapenemase-Producing Klebsiella pneumoniae"

_pharmaceuticals, 2024, doi:10.3390/ph17121683_

Round 1

Reviewer 1 Report

Comments and Suggestions for Authors

The manuscript by Strukova and colleagues reports on the development of an in vitro model for assessing the action of meropenem either alone or in combination with avibactam against three strains of Klebsiella pneumoniae. As experienced by others, and reflected in this manuscript, MIC determination for these combinations of drugs/pathogens is not trivial. Overall the experiments are clearly presented and I have only minor (mainly language based) comments. I feel strongly that the error bars should be included, despite the authors indicating that they might obscure the results. An alternate version of the figures with error bars should be included as a supplementary figure (at the least).

Specific comments

1.       I recommend that the authors or the MDPI handling team show future thoughtfulness to the reviewers by including continuous line numbering.

2.       The title should have the genus name in full

3.       P2, line 12. Please provide a reference for this factual statement after “resistance”

4.       P2, L14 (ESBLs)

5.       P3 Table 1 Iy may not be possible but getting the words merobactam (on on eline) and also avibactam, would be preferable in headers (as was possible in Table 2)

6.       P3 L6 Please define AUC on first use in main text

7.       More information on the strains would help the reader evaluate the ms. Are they hypervirulent, hyperviscous?

8.       Fig 1 and others. Please include error bars

9.       P6 Please include what the red circle means in the Fig legend for Fig 3. The reader should be able to interpret each Figure without having to read the later text.

10.   Abstract and elsewhere. The authors variably define PK as pharmacokinetic and pharmacokinetically. It should only have one definition.

11.   Materials 4.1 Please provide state for Hayward

12.   Methods 4.1 The particular bla genes should have KPC and Oxa as subscripts.

13.   P9 L3 Please abbreviate state to NJ

14.   P9 L7 inoculum

15.   P9 Please check that HFIM Is defined on first use.

16.   P9 immediately after HFIM, please use “An 8/1 ..”

17.   Section 4.3 middle.

18.   Section 4.3 middle; “ profiles in the in vitro…”

19.   Section 4.4 Has PD been defined?

20.   P10, L5 and L6 18-h culture….. 2-h period

21.   Title section 4.6 Analysis (to match capitalization of other headers)

22.   Section 4.6 Please use PD in second line

23.   Section 4.6. Please include versions with variation shown and modify text accordingly

24.   Refs 24 and 27 use different styles for PLoS One (ONE)

Reviewer 2 Report

Comments and Suggestions for Authors

Title: In vitro dynamic model evaluation of meropenem alone and in combination with avibactam against carbapenemase-producing K. pneumoniae

This manuscript is well-written by the authors. I do believe that if they can improve the manuscripts following all comments. It might have a chance to publish in the journal.

Comments

1. It would be better if the authors add the line number in the manuscript. It is easy to read and comment by reviewer.

2. Topic: Please re-write the topic.

3. Abstract: Please re-write or modify the following sentence “Carbapenem efficacy is declining due to increasing prevalence of Gram-negative bacteria that produce carbapenemases and are carbapenem resistant.”.

4. The present study focused on K. pneumoniae but there was no the information of K. pneumoniae or carbapenemase-producing K. pneumoniae in the abstract. Please add.

5. Abstract: Please modify the objective. It would be better if the authors use passive voice.

6. The authors should add the information of the results as quantitative data in the abstract.

7. Please add the results on dynamic model in the abstract.

8. Keywords: 5 keywords are enough.

9. Introduction: Please check and add the reference “Carbapenems are first-line treatments for critically ill infected patients”.

10. I suggest the authors to modify or re-write the introduction. What distinguishes your research between other? please give state of the art of your research

            -The first paragraph: describe the importance antimicrobial resistance (AMR) worldwide. 

            -The second paragraph: describe carbapenemase-producing K. pneumoniae.

-The third paragraph: describe the information meropenem and avibactam?

            -The fourth paragraph: describe the objective of this study. Why the authors are interested in this study?

11. Results: “The MICs varied 16-fold, ranging from 2 to 32 mg/L (Table 1)”. Please modify the sentence. The authors tested only 4 strains. Hence, the word “ranging from 2 to 32 mg/L” is not suitable.

12. Table 1 and Table 2 can be combined.

13. For time kill study (Figure 1), bacterial re-growth is observed after treatment with meropenem alone. Please discuss.

14. Discussion: Try to compare the results (the author’s hypothesis) with other finding by other researchers.

15. Please delete some introduction and result sentences in the discussion part.

16. Method: 4.1. Antimicrobial Agents and Bacterial Strains, please add the bacterial growth condition.

17. Method: 4.2. Susceptibility Testing: It should be “Antibiotic Susceptibility Testing”

18. Method: Please add the references for the drug combination. Actually, the antibacterial activity of the drug combination is investigated by Checkerboard assay to observe a synergistic effect. Why did the authors focus on this method? Please discuss.

19. Please remove some old references.  

Comments on the Quality of English Language

Please modify sentences and correct the grammar. 

Reviewer 3 Report

Comments and Suggestions for Authors

In this study, Strukova et al. evaluated the efficacy of a novel combination of meropenem with avibactam, a β-lactamase inhibitor. Overall, I am satisfied with the paper. However, I have several concerns that need to be addressed before publication:

1.         All the figure legends should indicate whether the figure shown is representative data and, if so, how many times the experiment was done.

2.         In their methods, the authors indicate that MIC values were obtained at least in triplicate, and the modal MICs were estimated. They should include a range of raw MIC values to give their readers a better sense of how their estimate corresponds to the data.

3.         In their methods, the authors write that ODs before and after 18 h incubation at 37 °C were estimated. Why were they estimated and not experimentally verified?

4.         In their methods, the authors write: The Inoculum effect was defined as an eightfold or greater increase in MIC when tested with HI compared to that for SI. What is the justification for this definition? Why 8X and not 7X or 9X?

5.         In their discussion, the authors write: Our study has a significant limitation: it did not include other β-lactam antibiotics or a wide range of K. pneumoniae strains; additional studies should be considered. Please elaborate. Which other K pneumoniae strains should they have included and how would this more diverse range of strains have impacted their data and the interpretation of their data.

Round 2

Reviewer 2 Report

Comments and Suggestions for Authors

The authors have edited and revised according to the reviewer comment.